# Massive Relief: Papillary Adenoma of the Lung in Asymptomatic Former Smoker Patient

**DOI:** 10.3390/diagnostics10110906

**Published:** 2020-11-06

**Authors:** Jelena Stojšić, Marko Popović, Federica Pezzuto, Jelena Marković

**Affiliations:** 1Department of Thoracic Pathology, Service of Pathology, University Clinical centre of Serbia, Pasterova 2, 11000 Belgrade, Serbia; jelenamark@live.com; 2Clinic of Thoracic Surgery, University Clinical Centre of Serbia, Koste Todorovića 26, 11000 Belgrade, Serbia; rekbucar@gmail.com; 3Department of Cardiac, Thoracic, Vascular Sciences and Public Health, Pathology Section, Medical School, University of Padova, 35121 Padova, Italy; federica.pezzuto@phd.unipd.it

**Keywords:** papillary tumor, papillary adenoma, peripheral tumor, lung, well-differentiated papillary mesothelioma, sclerosing hemangioma (pneumocytoma), alveolar adenoma

## Abstract

Benign epithelial tumors of the lung are uncommon and can represent a diagnostic challenge. Herein, we describe one such emblematic case. A 59-year-old former smoker male was admitted to the hospital complaining of cough for a long time. A radiological examination showed a centrally excavated mass strictly connected to the visceral pleura. The patient underwent tumorectomy. At gross examination, the tumor was composed of solid and cystic areas containing clear liquid. Histological examination highlighted a sub-pleural encapsulated tumor, with foci of capsular invasion, characterized by a single layer of columnar and cuboidal epithelial cells lining moderately cellular fibro-vascular cores. A wide spectrum of immunohistochemical markers was performed. The final diagnosis was suggestive of a peripheral pulmonary papillary tumor of undetermined malignant potential. At the last follow-up, six years after surgery, no recurrence or metastases were described. Reporting this case, we would like to point out the existence of these rare entities that should be taken into account in the diagnostic process, thus avoiding potential misdiagnosis. Moreover, the presence of capsular invasion should be better investigated in order to reconsider the exact terminology of the tumor and the classification of its malignant potential.

**Figure 1 diagnostics-10-00906-f001:**
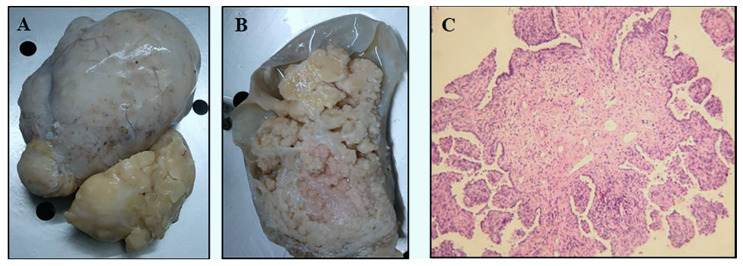
A 59-year-old former smoker male patient was admitted to the hospital for a persistent dry cough in the last few weeks. In the medical history, a treated hypertensive cardiomyopathy was reported. At chest X-Ray a nodule of 9 cm in maximum diameter was revealed. A chest CT scan was also performed, highlighting an excavated mass that arose from the posterior part of the lower right lobe visceral pleura. Radiological examination ruled out close and distant metastases. The patient underwent thoracotomy with tumorectomy without previous biopsy. At surgery, the tumor appeared encapsulated and was strictly connected to the visceral pleural, thus suggesting a mesothelial origin. At macroscopic examination, the tumor measured 9.5 × 7 × 4.5 cm and it was composed of papillary solid structures in the context of multicystic spaces with clear liquid content (**A**,**B**). Histologically, lung samples showed a subpleural lesion surrounded by a fibrous pseudocapsule. The neoplasia was characterized by a single layer of columnar and cuboidal epithelial cells lining moderately cellular fibrovascular cores, composed of plump, homogeneous spindle cells in a streaming fashion, and a moderate lymphocytic infiltration. The lining epithelial cells demonstrated minimal cytological and nuclear atypia, with no mitotic figures. Neither tertiary structures, as branching of papillary fronds without concomitant fibrovascular stromal, nor lepidic aspects were seen (**C**, haematoxylin and eosin, original magnification, ×100). Microfoci of tumor invasion in the pseudocapsular tissue and visceral pleura were described. The differential diagnosis included: sclerosing hemangioma, pulmonary papillary adenoma, papillary adenocarcinoma, and pseudoglandular carcinoid tumor. The diagnosis of pulmonary adenofibroma was also considered.

**Figure 2 diagnostics-10-00906-f002:**
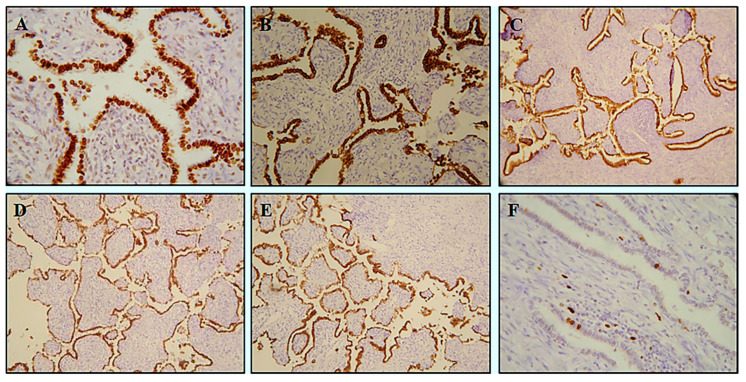
A large panel of antibodies was performed. The epithelial cells were positive for TTF-1 (clone 8G7G3/1, DAKO Cytomation, Glostrup, Denmark; dilution 1:100) (**A**), Napsin A (clone IP64, Novocastra^TM^ HD, Leica Biosystems, Newcastle, UK; dilution 1:400) (**B**), Surfactant B (Novacastra^TM^ HD Leica Biosystems, Newcastle, UK; dilution 1:50) (**C**), Cytokeratin 7 (clone OV-TL 12/30, DAKO Cytomation, Glostrup, Denmark; dilution 1:100) (**D**), pancytokeratin (clone AE1/AE3, DAKO Cytomation, Glostrup, Denmark; dilution 1:100) (MNF-116) (**E**) and negative for Synaptophysin (clone 27G1, Novocastra^TM^ HD, Leica Biosystems, Newcastle, UK; dilution 1:100) and Chromogranin A (clone 5H7, dilution Cytomation, Glostrup, Denmark; dilution 1:100). The neuroendocrine tumor differentiation was excluded. The Ki-67 (clone Ki-67P, Novocastra^TM^ HD, Leica Biosystems, Newcastle, UK; dilution 1:100) proliferative index was <1% (**F**), suggesting a low rate of growth and dismissing the hypothesis of a papillary adenocarcinoma. Pneumocytoma (so-called “sclerosing hemangioma”) was also ruled out based on the lack of sclerotic stroma, foamy histiocytes, and hemosiderin deposits and the overall absence of epithelial markers (TTF-1, Cytokeratin7, MNF-116 and Napsina A) in the underlying stroma. Likewise, the abundance of papillary structures and the features of the stroma dismissed the diagnosis of a pulmonary adenofibroma. The final diagnosis was in favor of a “peripheral pulmonary papillary tumor”. The invasive behavior of the tumor cells with foci of invasiveness towards the pericapsular tissue suggested the definition of “undetermined malignant potential”. At the last follow up, six years after surgery, no recurrence or metastases were reported in this patient.

Pulmonary papillary adenoma is a rare neoplasm, predominantly occurring in peripheral lung areas, usually asymptomatic, that consists of cuboidal to columnar cells without nuclear pleomorphism and absent or low proliferative index. It mainly affects the male population, with an age range from 2 months to 70 years [1]. Pulmonary papillary adenomas of indeterminate malignant potential are described only in a few reports [2,3,4]. In these cases, although the clinical behavior has been demonstrated as benign, microfoci of invasion in the pseudocapsular tissue and visceral pleura suggested a local aggressiveness or a potential malignant behavior. More recently, a malignant transformation has also been reported, with the features of acinar and micropapillary adenocarcinoma within the mass [5]. Some differential diagnoses should be taken into consideration, such as differentiated papillary mesothelioma, sclerosing hemangioma (pneumocytoma), alveolar adenoma, and lung adenocarcinoma. Immunohistochemistry could be helpful in distinguishing these entities [6,7,8,9,10]. Depending on the localization and size of this tumor, surgical resection (wedge resection or lobectomy) is always the gold standard treatment [2,11]. The description of infiltrative borders, as also highlighted in our case, supports the need of reconsidering the definition of the tumor, replacing the term “pulmonary papillary adenoma” with “peripheral papillary tumor of undetermined malignant potential”. Further molecular investigations are mandatory to explore if this entity could be a step involved in the development of malignant forms.

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
