# Peer review of "Massive Relief: Papillary Adenoma of the Lung in Asymptomatic Former Smoker Patient"

_diagnostics, 2020, doi:10.3390/diagnostics10110906_

Round 1
Reviewer 1 Report
Stojsic et al. provide a nice and interesting case report about a rare pulmonary tumor. I have only a couple of minor Points to raise:
- did the Patient undergo "tumorectomy" (line 39) without previous biopsy or radiological staging to rle out distant metastases?
- I suggest also adding a CT Image of the tumor
- some minor spelling mistakes: line 27 investigated, line 35 persistent dry coug, line 36 hypertensive cardiomyopathy
- mention the rarity of the Tumor also in the discussion (lines 72 ff).
- maybe make the title more concrete. One somewhat playful example could be "Massive relief: papillary adenoma of the lung in a symptomatic ex smoker"
Author Response
Dear Reviewer,
the authors appreciate the time invested by you in the review of this paper.
- Patient went under tumourectomy without previous biopsy and radiological staging to rule out the eventual metastases.
- Unfortunately, there was no CT-scan to obtain as previously discussed.
- All mentioned spelling mistakes as well as a couple more were corrected.
- Rarity of the tumour was added in discussion.
- The title suggestion was gladly accepted.
Kind regards,
dr Stojsic and co-authors
Reviewer 2 Report
This is nice case presentation and clear description of an unusual tumour in a previously fit smoker. I agree with the authors, it’s important to report these rarer cases for the purposes of a fuller differential diagnosis.
This submission would be greatly improved with the inclusion of the diagnostic radiology - presenting CXR / CT thorax.
Author Response
Dear Reviewer,
the authors appreciate your time invested in the review of our manuscript.
Unfortunately, thorax CT-scan was not performed, and CXR was lost by the patient himself without saving in hospital database.
Kind regards,
dr Stojsic and co-authors